# Influence of the Natural Zeolite Particle Size Toward the Ammonia Adsorption Activity in Ceramic Hollow Fiber Membrane

**DOI:** 10.3390/membranes10040063

**Published:** 2020-04-04

**Authors:** Mohd Ridhwan Adam, Mohd Hafiz Dzarfan Othman, Siti Hamimah Sheikh Abdul Kadir, Mohd Nazri Mohd Sokri, Zhong Sheng Tai, Yuji Iwamoto, Masaki Tanemura, Sawao Honda, Mohd Hafiz Puteh, Mukhlis A. Rahman, Juhana Jaafar

**Affiliations:** 1Advanced Membrane Technology Research Centre (AMTEC), School of Chemical and Energy Engineering (SCEE), Faculty of Engineering, Universiti Teknologi Malaysia, UTM, Skudai, Johor 81310, Malaysia; ridhwan_adam@yahoo.com (M.R.A.); nazri@petroleum.utm.my (M.N.M.S.); taizhongsheng92@gmail.com (Z.S.T.); mukhlis@petroleum.utm.my (M.A.R.); juhana@petroleum.utm.my (J.J.); 2Institute of Medical Molecular Biotechnology, Faculty of Medicine, Sungai Buloh Campus, Universiti Teknologi MARA (UiTM), Jalan Hospital, Sungai Buloh, Selangor 47000, Malaysia; 3Department of Environmental and Materials Engineering, Nagoya Institute of Technology, Gakiso-cho, Showa-ku, Nagoya 466-8555, Japan; iwamoto.yuji@nitech.ac.jp (Y.I.); honda@nitech.ac.jp (S.H.); 4Department of Physical Science and Engineering, Nagoya Institute of Technology, Gakiso-cho, Showa-ku, Nagoya 466-8555, Japan; tanemura@system.nitech.ac.jp; 5Department of Environment, School of Civil Engineering (SCE), Faculty of Engineering, Universiti Teknologi Malaysia, UTM, Skudai, Johor 81310, Malaysia; mhafizputeh@utm.my

**Keywords:** natural zeolite, adsorptive ceramic membrane, phase inversion, particle size, adsorption

## Abstract

Natural zeolite is widely used in removing ammonia via adsorption process because of its superior ion-exchange properties. Ceramic particle size affects the adsorptivity of particles toward ammonia. In this study, hollow fiber ceramic membrane (HFCM) was fabricated from natural zeolite via phase inversion. The effect of natural zeolite particle size toward the properties and performance of HFCM was evaluated. The results show that the HFCM with smaller particle sizes exhibited a more compact morphological structure with better mechanical strength. The adsorption performance of HFCM was significantly improved with smaller particle sizes because of longer residence time, as proven by the lower water permeability. A high adsorption performance of 96.67% was achieved for HFCM with the smallest particle size (36 μm). These findings provide a new perspective on the promising properties of the natural zeolite-derived HFCM for ammonia removal.

## 1. Introduction

Potable water scarceness has become a crucial problem across the globe for many years. The introduction of many contaminants into water bodies has worsened the effect of water pollution. The presence of contaminants, such as heavy metals, toxic chemicals, and industrial wastes, is known as one of the foremost causes of this problem [1]. The rapid growth of industrialization, along with the increasing human population, has brought a massive impact on drinkable water demand. Thus, the necessity to treat wastewater for a sustainable potable water source is an inevitable challenge [2]. The present water purification and wastewater treatment are no longer adequate to meet the needs of the future generation.

Ammonia is one of the known water pollutants. It is a common contaminant in both municipal and industrial wastewaters. The growth in industrial processes specifically in coking, coal gasification, and petroleum refining; and industrial productions of chemical fertilizers, pharmaceuticals, and catalysts; has led to a large production of ammoniacal wastewater [3]. Excessive nitrogen compound in wastewater is a significant pollution burden since it has distinctive characteristics and leads to the depletion of dissolved oxygen required for aquatic life. Moreover, excessive nitrogen compound has toxic effects on fish, lowers disinfection efficiency, and accelerates the corrosion of metals and construction materials [4]. 

One of the best-known solutions for tackling this issue is the separation and purification of wastewater. This solution requires the implementation of cutting-edge equipment, such as membrane separation technologies. Up to now, membrane technology has been utilized in a broad range of applications to separate individual components from either liquid or gas mixtures. Past studies have shown that membrane technology is applicable for removing ammonia in wastewater in a clean and effective way. This could be ascribed by the high ammonia removal efficiency in addition to the compact design and low energy cost compared to conventional ammonia wastewater treatments such as air-stripping, denitrification, coagulation and flocculation, and biological treatment. 

Currently, most of the commercially available membranes in the market are polymeric-based membranes, which are cheaper to produce and have high perm selectivity [5,6,7]. However, these polymeric membranes are limited to mild operating conditions, specifically low operating temperature and pressure. This could be attributed by their weak thermal stability and ease of fouling [8]. Membrane fouling is a severe problem, and further treatment is needed to ensure excellent membrane performance [9]. Strong hydrophilicity helps in reducing membrane fouling. Therefore, researchers are now focusing on the development of ceramic-based membrane as it has high durability in harsh environments and possesses high hydrophilicity [10]. Additionally, most of the membranes used in treating wastewater containing contaminants such as heavy metals, humic acid, pharmaceutical wastes, and ammonia are mainly of inorganic ceramic materials. These materials possess outstanding durability under high pressure and temperature, great chemical stability, good defouling property, and long-lasting usability [11]. However, the needs of high sintering temperature and expensive starting materials such as alumina, zirconia, and titania have led to the least choice of ceramic materials to be used as membrane [12,13]. As a consequence, there is a necessity to find inexpensive ceramic materials to replace costly ceramics.

There is an eminent need to search for alternative membrane materials. Various studies have been devoted to fabricating the ceramic membranes using minerals and waste materials such as clay, fly ash, bauxite, and kaolin [14,15,16]. However, these materials can be costly, are less effective for ammonia adsorption, and possess inconsistency potentially due to different batches of the waste [17]. For example, bauxite contains a high amount of alumina; thus, it requires high sintering temperature to be produced as a ceramic membrane [18]. Meanwhile, the usage of the fly ash may lead to the production of porous materials that contain various phases including mullite, anorthite, and cordierite [19]. The membranes of pure fly ash typically possess low mechanical strength and need additives to enhance their durability [20]. 

Alternatively, natural zeolite has been studied as an alternative material to be developed as a low-cost ceramic membrane. Natural zeolite is abundantly found around the world as crystalline hydrated aluminosilicates of alkali and earth metals [21]. Dong et al. fabricated a tubular microfiltration membrane for water filtration [22]. Although this approach is interesting, the big configuration of the tubular membrane resulted in low performance because of low surface area [23]. As an alternative, hollow fiber configuration offers several advantages such as large membrane area per unit volume, good mechanical support to withstand liquid separation or backwashing, and easy handling of fabrication and operational processes [24,25]. These properties will improve the separation performance of the fabricated ceramic membrane. Thus, this study evaluated the potential of a natural zeolite (clinoptilolite) as a microfiltration membrane for the removal of ammonia via adsorption. Also, the excellent adsorbent and separation properties of this natural zeolite have made it a cheaper ceramic material for membrane fabrication. Additionally, this natural zeolite guarantees promising results in the removal of ammonia in water [26]. 

To the best of our knowledge, no similar studies have been reported the fabrication of a natural zeolite-based adsorptive hollow fiber ceramic membrane (HFCM) for the removal of ammonia in water. A study was conducted on natural zeolite in HFCM for the removal of chromium(VI) from water [27]. Although the adsorptive membrane produced is similar, the application of the study was limited for the treatment of chromium(VI), and the effect of particle size on the membrane performance was not investigated. Particle size significantly affects membrane properties in terms of mechanical strength, porosity, and water permeability [28]. In addition, a small particle size offers a high surface area, resulting in a large number of active sites [29]. This property eventually increases the adsorptivity of the membrane in ammonia removal. Thus, this work reports the effect of natural zeolite particle size on the physical and chemical properties of the adsorptive HFCM for the treatment of ammonia wastewater. The effects of the ceramic particle size on the physicochemical properties such as membrane compactness, crystallinity of the ceramic, and microtopography of the membrane as well as the performance of the HFCM were studied in detail in this work specifically for the adsorptive removal of ammonia in water treatment. The adsorptive performance is the major concern in this study and the physicochemical properties were also taken into account. The changes in the physical properties of the HFCM eventually affected the membrane performance.

## 2. Experimental

### 2.1. Materials

The materials used for the fabrication of the adsorptive HFCM were natural zeolite (Shijiazhuang Mining Trade Co. Ltd. Ziaoning, China), Arlacel P135 (polyethyleneglycol-30 dipolyhydroxystearate, Uniqema, East Yorkshire, UK) as the dispersant, N-methyl-2-pyrrolidone (NMP, AR grade, QRëC™, Selangor, Malaysia) as the solvent, and polyethersulfone (PESf, Radal A300, Ameco Performance, Greenville, SC, USA) as the polymer binder. All chemicals were purchased and used without any further purification.

### 2.2. Membrane Fabrication and Characterisation

Natural zeolite with different particle sizes (36, 50, and 75 µm) was ground and sieved. The membrane dope suspension was prepared by dissolving Arlacel P135 dispersant into NMP before the addition of pre-dried natural zeolite powder (45 wt % for each particle size suspension). The suspension was then subjected to ball milling process (NQM-2 planetary ball mill) for 48 h. Then, PESf (5 wt %) was added into the suspension, and the mixture was further milled for another 48 h. Before the extrusion of the suspension using phase inversion technique, the dope suspension was degassed for 30 min at room temperature to eliminate the trapped air in the suspension to prevent defect in the pore formation in the membrane structure. After degassing, the spinning suspension was introduced into a syringe pump and extruded through a tube-in-orifice spinneret. Tap water was used as the internal coagulant at 15 mL/min. The fiber membrane green bodies that passed through a 5 cm air-gap distance were immersed in a water coagulant bath for 24 h to allow the completion of the phase inversion process. Afterward, the membrane precursor green bodies were dried at room temperature. Finally, the fiber precursors were sintered in air for 4 h at 1050 °C, as the best HFCM is obtained when sintered at this temperature [30]. The heating rate was 2 °C/min throughout the heating and cooling process of sintering.

The viscosity of the spinning suspension was measured promptly before the spinning process using a viscometer (Brookfield model DV1, Middleboro, MA, USA) at a shear rate range between 1 and 100 s^−1^ at room temperature. The morphologies of each membrane before and after the surface modification process were examined using scanning electron microscopy (SEM, Hitachi model TM 3000, Tokyo, Japan) at different magnifications. Before the SEM analysis, each membrane was coated with gold for 3 min under vacuum. This analysis consisted of both surface and cross-sectional anatomy of the membranes. The microtopography of the natural zeolite and fabricated HFCM was analyzed using transmittance electron microscopy (TEM, JEOL JEM-2100, Tokyo, Japan). The samples were directly mounted on the TEM sample holder without any additional treatment. TEM analysis was performed with a vacuum chamber pressure of less than 2.5 × 10^−5^ Pa. The crystal phase of the natural zeolite as raw material compared to that of zeolite from HFCM upon the sintering process was analyzed using X-ray diffraction analysis (XRD, Shimadzu model XD-D1, Kyoto, Japan). The mechanical strength of the membrane was measured using a three-point bending machine (Instron model 3342, Norwood, MA, USA). The flexural strength of the membrane was calculated using the following equation:(1)$$\sigma F=8FLD_{o}/\pi (D_{o}^{4}-D_{i}^{4})$$
where *F* is the maximum load at which the fracture occurred while *L*, *D_o_*, and *D_i_* are the length of span (43 mm), the outer diameter, and the inner diameter of the hollow fibers, respectively.

Mercury porosimeter (AutoPore IV, 9500, Micromeritics, Norcross, GA, USA) combined with Micromeritics software (version 1.09) was used to estimate the pore size distribution and overall porosity of the fabricated membranes. Water permeation was measured using a crossflow membrane permeation system, as illustrated in Figure 1. Water permeation flux (*F*, L/m^2^∙h) was calculated using Equation (2) from the measured volume of the permeate (*V*, L), membrane area (*A*, m^2^), and time (*t*, h).
(2)$$F=\frac{V}{At}$$

### 2.3. Ammonia Removal by Adsorptive Natural Zeolite HFCM

The ammonia removal performance of the HFCM via adsorption was evaluated using a dead-end water permeation setup with different grades of synthetic ammonia wastewater. The performance was measured using HFCM of different particle sizes in 50 mg/L and pH 7 ammonia feed solution. The ammonia contents in both feed and permeate solutions for each analysis were determined using UV–visible spectrophotometry (Hach model DR 5000, Guelph, ON, Canada) with the aid of ammoniacal reagent kit containing ammonium salicylate and ammonium cyanurate as the color indicator for the ammonia content. The color intensities were measured in the mode of absorbance and translated into ammonia content by comparing them to the calibration curve prepared before the spectrophotometric analyses. The ammonia removal performance was calculated using the following equation:(3)$$R=\frac{C_{f}-C_{p}}{C_{f}}\times 100$$
where *C_f_*
_and_
*C_p_* are the ammonia concentration in the feed and permeate, respectively. 

## 3. Results and Discussion

### 3.1. Natural Zeolite HFCM Fabrication and Characterisation

The characterization and the effect of the particle size toward the physicochemical properties of the zeolite adsorptivity on the ammonia were studied in detail. The common analysis of the particle size in adsorption activities is normally reported in the powder suspension form of adsorbent [29,31]. The analysis of particle size effect incorporating with the membrane configuration is the novelty of this current work. Although there are studies that report the effect of particle size on membrane technology, they are limited to the particle size of the additive materials embedded in the membrane matrix [32,33]. This current work highlights a new perspective on the effect of particle size on the ceramic membrane adsorption performance. 

#### 3.1.1. Rheology of the Ceramic Suspension

Viscosity of the dope suspension plays an important role in the determination of the membrane structure. It affects the phase inversion process by controlling the formation of sponge- or finger-like structures in the membrane. Figure 2 depicts the rheology of the zeolite dope suspension prepared using different particle sizes of zeolite powder. The smaller particle size of ceramic led to a highly viscous dope suspension. Thus, the high viscosity led to the formation of the sponge-like structure. In addition, the graph shows the correlation between particle size and viscosity of the dope suspension. The smaller zeolite powder particles dispersed well in the suspension and thus thickened the suspension. A similar trend of findings was reported on the dispersion of nanoparticles of different sizes in a polymeric suspension [34]. Similarly, the increased particle size in this study reduced the specific surface area of the particle, providing a poor contact between the particles. In contrast, 50 µm particle size gave a thinner suspension, which was less viscous compared to that of the 36 µm zeolite particle. The least viscous suspension using 75 µm zeolite powder was obtained.

According to Kingsbury and Li, the viscosity of dope suspension mainly dominates the formation of finger-like structure on the HFCM [35]. Their study showed that the formation of finger-like structure was more favorable, obtained from the use of the less viscous dope suspension. Kingsbury and Li also proposed the threshold value of the alumina dope suspension viscosity at 12.1 Pa∙s at the shear rate of 30 s^−1^. The study also revealed that a viscosity exceeding this threshold value is less likely to produce a finger-like structure and, in turn, would be more dominated by the sponge-like structure formation across the membrane.

In this study, the viscosities of the suspension at the shear rate of 30 s^−1^ were 0.6880, 0.6302, and 0.4558 Pa∙s for 36, 50, and 75 µm zeolite, respectively. The decreasing trend of dope suspension viscosities as the particle size of zeolite increased resulted in the formation of more finger-like and/or macrovoid structures in the membrane. The less formation of finger-like structures attained in the membrane fabricated using 36 µm zeolite particle indicates that this dope suspension was approaching the threshold value of the zeolite suspension. In other words, the formation of the sponge-like structure is more likely to be attained if the viscosity of the dope suspension is more than 0.6880 Pa∙s. Furthermore, the formation of sponge-like instead of finger-like structure in a thicker suspension is due to the demixing process between solvent and nonsolvent during the phase inversion process. A similar finding was obtained by Kingsbury et al. on the dope suspension thickening effect toward the control of finger- and sponge-like structures across the membranes [36]. 

#### 3.1.2. Morphological Behavior of the HFCM

The morphology of the HFCM spun using different grades of natural zeolite particle sizes is depicted in Figure 3. The SEM micrographs of the HFCM were taken at different magnifications. The particle size plays an important role in determining the morphological structures of the membrane. As discussed in the previous section, the HFCM structure is mainly affected by the viscosity of the ceramic dope suspension. The HFCM spun by the most viscous dope suspension (36 μm of zeolite powder) resulted in the densest membrane structure and the thickest membrane wall. The highly viscous dope suspension, despite the same ceramic loading and sintering temperature, produced a more packed membrane. The finer ceramic particles were closely packed in a uniform arrangement, and thus the formation of pore or voids in between the ceramic particles was reduced.

Apart from the cross-sectional morphology of the HFCM, the SEM analysis was also done to evaluate the inner and outer surfaces of the HFCMs (Figure 4). The outer surface of the HFCM lumen was rougher than the inner surface. The particle distribution and arrangement on the membrane surface were highly influenced by the bore fluid effect during the phase inversion process. During the process, the internal coagulant that acted as the lumen former for the HFCM has indirectly arranged the particle of the zeolite uniformly. The hydrodynamic force of the bore fluid eventually thrusted the dope suspension as well as the ceramic particle to a more packed arrangement, resulting in a smooth surface [37]. On the other hand, the rough surface of the outer contour of the HFCM might be due to the weak hydrodynamic force of the external coagulant, reducing the solidification rate of the dope suspension at the outer surface. Additionally, the inner surface of the HFCM showed cracking upon the increment in the particle size of the natural zeolite powder. This phenomenon could be attributed to the thinning effect of the dope suspension when the bigger ceramic particles were used. The high hydrodynamic force thrusted the ceramic particles within the thin dope suspension, thus promoting the crack formation. This defect also weakened the membrane strength. The arrangement of the ceramic particles on the outer surface was loose and less packed compared to that on the internal surface. This occurrence is important because it determines the effectiveness of the adsorption process, which mainly depends on the surface area of the adsorbent. A bigger surface area possesses a better adsorption performance. This bigger surface area normally comes from the rough surface. Figure 5 illustrates the phase inversion process that contributed to the formation of the smooth inner surface and the rough outer surface of the HFCM.

The microtopography of the natural zeolite powder and the HFCM derived from the natural zeolite (upon the sintering process) was analyzed using TEM and is depicted in Figure 6. The HFCM spun using the 36 μm zeolite powder was selected for this analysis. The raw natural zeolite had a lamellar structure. The natural zeolite edges were sharp with more tubular particles. Upon the sintering process, the natural zeolite edges melted, resulting in a smoother, thinner, and more rounded shape compared to the unsintered natural zeolite powder. This change may affect the membrane performance as the surface area of the HFCM was reduced because of the heat treatment. A similar observation was reported on the modification of the natural zeolite using acid soaking where the edge of the zeolite particles softened [38]. The smoother surface offers a limited surface area/volume compared to the rougher surface and is expected to give better adsorption toward the ammonia removal. However, the adsorption/filtration process does not only rely on the surface area for better adsorption efficiency. 

#### 3.1.3. Crystallinity Properties of the HFCM

The XRD patterns of the natural zeolite and HFCM are depicted in Figure 7. The XRD analysis of the natural zeolite was validated by the diffraction peaks at 2θ of 9.768°, 11.105°, 13.220°, 16.796°, 18.894°, 20.762°, 22.247°, 22.615°, 25.930°, 26.547°, 28.040°, 29.885°, 31.861°, 32.580°, 36.451°, and 50.051° and was confirmed by the standard pattern of the clinoptilolite (01-079-1460 JCPDS card). This pattern of peaks indicates that the main phase of this natural zeolite is clinoptilolite. The same trend of finding was reported in the literature [39]. Additionally, a slight amount of quartz and cristobalite phases were found in the XRD pattern of this natural zeolite sample. It is believed that these phases exist due to the presence of impurities. Upon the sintering process, the crystallinity of the natural zeolite was changed because of the heat treatment. The comparative analysis between these two peaks revealed less intense peaks for the heat-treated sample of the HFCM. In addition, some amorphous structures formed in the HFCM sample upon the sintering process. This was confirmed by the disappearance of some peaks on the diffractogram. The disappearance of major peaks of clinoptilolite was observed in all spectra of the HFCM. Additionally, the spectra of all HFCMs showed that the major phases present were quartz (01-083-2187 JCPDS card), cristobalite (01-077-8629 JCPDS card), and anorthoclase (01-077-8526 JCPDS card). The flattening of peaks of natural zeolite upon the sintering process was due to the burn-off process of the elements. Hence, the remaining peaks belong to the crystal phase with a high melting point. Similar findings were reported when different treatments were done on the natural zeolite sample [40,41]. As mentioned earlier in the previous section, the heat treatment reduced the surface area of the natural zeolite because of the grain growth phenomenon that merged the natural zeolite particles toward the end of the process [42]. On the other hand, the particle size plays no significant role in the changes in the crystal phase of the HFCM. This is proven by the presence of all phase peaks in all diffractograms.

#### 3.1.4. Mechanical Strength and Water Permeability of the HFCM

Apart from the physicochemical properties of the morphological and crystallinity behaviors of the produced HFCM, the mechanical strength of the HFCM was significantly influenced by the particle size of the natural zeolite used in the fabrication step. Figure 8a depicts the mechanical strength of each HFCM produced by different grades of the zeolite particle size. The mechanical strength of the HFCM significantly decreased with the increment in the particle size of the natural zeolites. The mechanical strength of the HFCM with 36 μm zeolite of 52.92 MPa decreased to more than half upon the increment in zeolite powder size to 75 μm. The value of the mechanical strength of the HFCM in this work is comparable to those reported in other studies, although the sintering temperature in this study is much lower [37,43]. The reduction in the HFCM mechanical strength could possibly be due to the particle arrangement in the structure. The smaller particles were aligned in a more uniform structure with a closely packed and dense assemble. This phenomenon is clearly seen from the cross-sectional micrograph of each membrane (Figure 3). Besides the compact structure, the membrane produced possessed less gap or space between the particles. The bigger space or gap between the particles may lead to defect formation in the structure, reducing the mechanical strength of the HFCM. Likewise, this concept of structural densification shows some similarity to the effect of sintering temperature [37]. In the sintering process, the ceramic membrane structure is affected by the grain growth phenomenon. The higher the temperature, the better the mechanical strength due to the densified structure with closely packed ceramic particles. This phenomenon could be imitated by the particle size effect. The smaller particles tend to form a denser membrane, increasing its strength. 

Another physical property that determines the effectiveness of the ceramic membrane is water permeability. Figure 8b shows the water permeation profiles of the HFCMs fabricated using different grades of natural zeolite particle size at different pressures. The water permeability increased with the increment in natural zeolite particle size. The lowest water permeability was recorded in the 36 μm zeolite HFCM of 228.25 L/m^2^∙h at 1 bar. At the same pressure, the water permeability significantly increased to 687.93 L/m^2^h when the membrane was changed to the 75-μm zeolite HFCM. Upon the increment in the water pressure, each HFCM possessed a significantly increased water permeation with the 75-μm zeolite HFCM recording the highest permeability of 2589.84 L/m^2^h at 3 bar. In comparison, the lowest was 601.06 L/m^2^h, achieved by the 36-μm zeolite HFCM under the same condition. These findings show that the water permeability of the HFCM is mainly determined by the morphology of the membrane.

Similar to the mechanical strength, the compactness of the ceramic particles in the membrane structure mainly affected the permeability of the HFCM. The more compact the membrane structure (mainly achieved by the smaller natural zeolite particles), the lower the water permeability. The effect of particle size had been studied and reported [28]. The effect of ceramic particle size was almost similar to that observed in this study.

The permeability and compactness of the HFCM are further supported by the porosity data obtained via the mercury intrusion porosimetry analysis. Figure 9a depicts the pore size distribution of the HFCMs derived from different ceramic particle sizes. Different particle sizes did not merely affect the pore size distribution of the membrane. The same range of the membrane pore size between 2.8 and 10.3 μm were not changed for all the membranes from different particle sizes. A similar trend of finding was reported in another study [44]. The single broad peak possessed by each membrane indicates that the membranes were composed of symmetrical sponge-like structure, represented by uniform pore formation throughout the membrane. Additionally, the mercury intrusion intensity decreased upon the reduction of the ceramic particle size. This indicates that the pore size became smaller, which is attributable to the smaller particle size. Othman et al. reported the same findings in their study [45]. In addition, the slight mercury intrusion in the smaller range of pore size (0.18–1.5 μm) indicates the grain boundaries of the natural zeolite particles contained in the membrane [46].

In addition, the mercury intrusion porosimetry analysis also signifies the porosity degree of the produced membrane. Figure 9b shows that the increment in particle size increased the porosity of the membranes produced with porosity degrees of more than 45%, which indicates that the produced membranes were composed of relatively large pores that were sufficiently porous for a ceramic membrane [47]. The higher degree of membrane porosity is crucial for the determination of higher water permeability, where it is highly required for the water treatment process membranes [48]. These findings reveal that the membrane fabricated using larger particle size possessed a higher water permeability, resulting in the poorest ammonia uptake through the adsorption process.

### 3.2. Ammonia Removal by Natural Zeolite HFCM

The adsorptive performance of the HFCM toward ammonia removal was evaluated using the adsorptive HFCM derived from different grades of natural zeolite particle size. Figure 10 depicts the percentage of removal and water permeability of the 50 mg/L ammonia feed solution at 1 bar and pH 7. The pressure was chosen because of the lowest permeability produced, increasing the residence time acquired for a better adsorption process [49]. The membrane with the smallest particle size (36 μm) showed the highest ammonia removal of 96.67%. The high performance of ammonia rejection could be attributed to the low water permeability of the feed solution (201.54 L/m^2^h). A low water permeation leads to high adsorption because of the long residence time between the adsorbate and adsorbent. According to Foo et al. the adsorption rate increases rapidly at the early stage, ascribed to the readily accessible active sites on the adsorbent [49]. Upon prolonged contact, the adsorbate uptake becomes less efficient because access to active sites is limited. Also, the reduced adsorption performance could possibly be due to insufficient residence time for the adsorbate within the adsorptive membrane at high permeation, because the ammonia solution left the membrane before equilibrium was attained [50]. Thus, the 75-μm zeolite HFCM, which possessed the highest water permeability, yielded the lowest ammonia uptake. 

Figure 11 illustrates the interaction of the membrane particle compactness (contributed by the particle size) and its water permeability and adsorption performances. The more compact membranes, attained by the smallest particle, retain (delays) more water to pass through the membrane that eventually prolongs the retention time for an effective adsorption process. Apart from the interparticle interaction, the adsorption of ammonia can also be attributed to the intraparticle pore structure. Figure 12 depicts the schematic of the intraparticle pores of an adsorbent for the adsorption process [51]. The presence of pores in an adsorbent particle increases the adsorption surface that serves as the active site for the adsorption to occur [52]. Besides promoting a more compact membrane structure, smaller natural zeolite particles also offer more surface area (including the intraparticle pores, namely macropore, mesopore, and micropore) that finally increases the adsorption capacity of the fabricated membranes.

In addition, the efficiency of the natural zeolite clinoptilolite in adsorbing ammonia showed a good performance where up to 96.67% of ammonia was successfully removed from the feed solution. This excellent behavior indicates that natural zeolite clinoptilolite possesses a great potential to be developed as an adsorptive membrane. Table 1 lists the ammonia removal efficiencies of several natural zeolites, particularly clinoptilolite, in the form of powder suspension compared to this study (adsorptive membrane). The findings show that the ammonia removal in this study is compatible with that of natural zeolites reported in the literature. Therefore, the ammonia adsorption intention for this study is highly achievable and practical.

On the other hand, this study serves as a new perspective on the fabrication of low-cost ceramic membranes for various applications. Table 2 summarizes the fabrication of low-cost membranes derived from various materials and targeted for different applications. Low-cost membranes have been widely used for many applications. Of all the applications, these membranes are mainly used for water treatment processes, including oil–water separation, dye removal, suspended solid filtration, and heavy metal elimination. Therefore, all these studies have revealed that low-cost membrane fabricated from natural raw materials and waste products are capable of achieving high performance in treating the wastewater from various industries. 

## 4. Conclusions

This paper presents the characteristics study as well as the feasibility of HFCM derived from the natural zeolite for the adsorption of ammonia from wastewater. Different grades of zeolite particle size were used to fabricate the HFCM and extruded at fixed membrane fabrication parameters. The physicochemical properties of the HFCM were thoroughly examined and observed throughout this study. It is concluded that the particle size plays an important role in determining the properties of the membrane and thus controls its performance. The zeolite with 36 μm particle size possesses the most promising role in producing the best HFCM accompanied with the most favorable physical properties, namely morphology, mechanical strength, and water permeability. These properties are vital in the adsorptive performance of the ammonia removal. The compact and uniform particles in the membrane structure assured the acceptable strength for the experimental handling. In addition, this feature has an added value in slowing down the water permeability of the membrane, ensuring high performance for the adsorption process with more than 96% of ammonia removal. Additionally, the best possible sintering temperature aided the formation of the compact HFCM through the grain growth process along with the membrane densification process. This research will certainly provide a better understanding of the fabrication of effective adsorptive HFCM with controlled morphology, mechanical strength, and water permeability to improve the adsorption capability for ammonia removal.

## Figures and Tables

**Figure 1 membranes-10-00063-f001:**
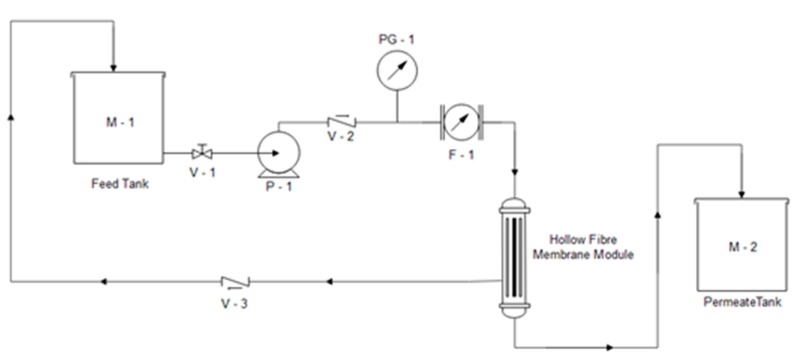
Schematic diagram of the adsorptive hollow fiber ceramic membrane (HFCM) system setup [30].

**Figure 2 membranes-10-00063-f002:**
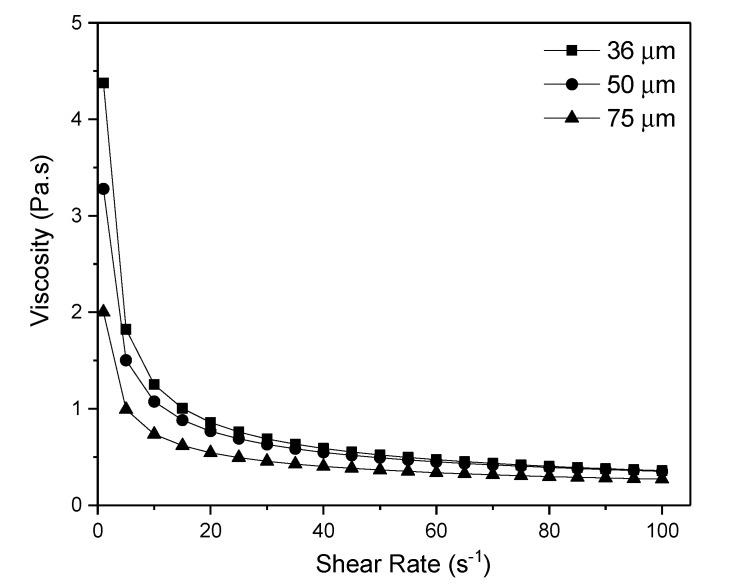
Rheological profile of suspension containing zeolite/NMP/PESf/Arlacel with different sizes of zeolite particle.

**Figure 3 membranes-10-00063-f003:**
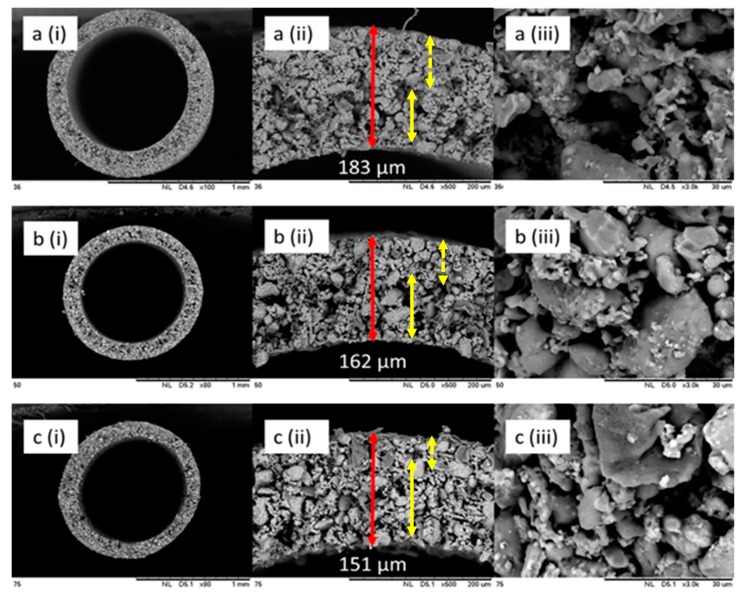
Scanning electron microscopy (SEM) micrographs of the HFCMs at different magnifications (**i**) ×60, (**ii**) ×500 and (**iii**) ×3000 of (**a**) 36 μm, (**b**) 50 μm, and (**c**) 75 μm, spun at 45 wt % ceramic loading, 5 cm of air-gap distance and 15 mL/min of bore fluid flowrate. (Note: full yellow arrow noted the finger-like or macro void region while dotted arrow showed the sponge-like structure).

**Figure 4 membranes-10-00063-f004:**
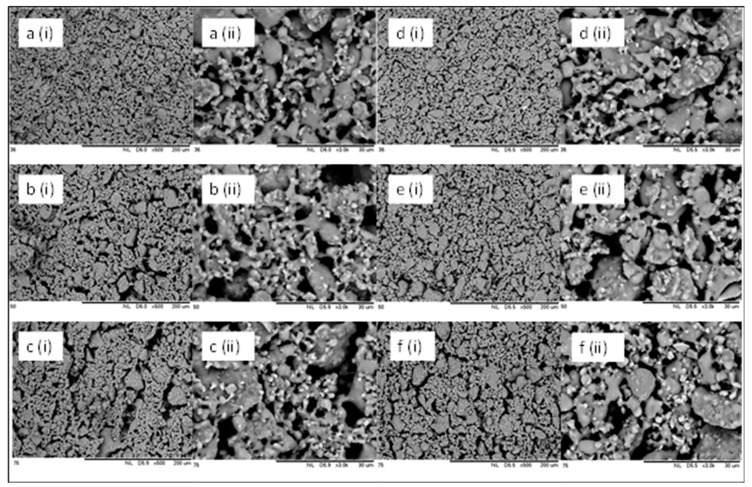
SEM micrographs of the HFCMs of inner surface of (**a**) 36 μm, (**b**) 50 μm, and (**c**) 75 μm, and outer surface of (**d**) 36 μm, (**e**) 50 μm, and (**f**) 75 μm at different magnifications of (**i**) ×60 and (**ii**) ×500.

**Figure 5 membranes-10-00063-f005:**
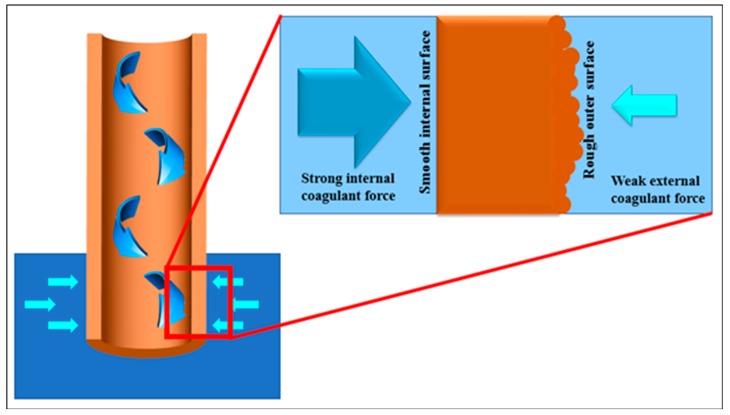
The illustration of the phase inversion process that influences the physical appearance of the inner and outer surfaces of the HFCM.

**Figure 6 membranes-10-00063-f006:**
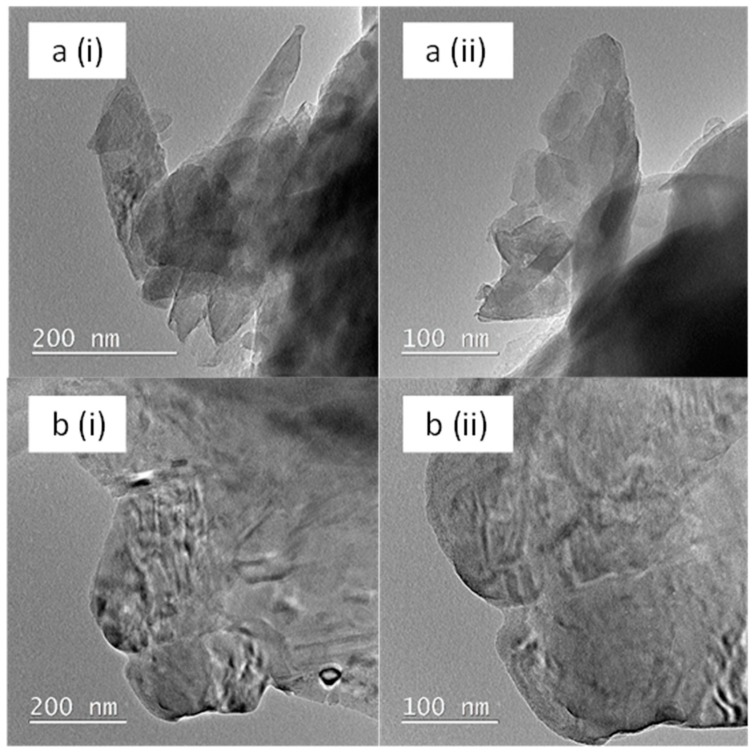
The transmittance electron microscopy (TEM) microtopography of the (**a**) natural zeolite powder and (**b**) HFCM derived from the 36 μm natural zeolite, at different magnifications of (i) ×60 and (ii) ×500.

**Figure 7 membranes-10-00063-f007:**
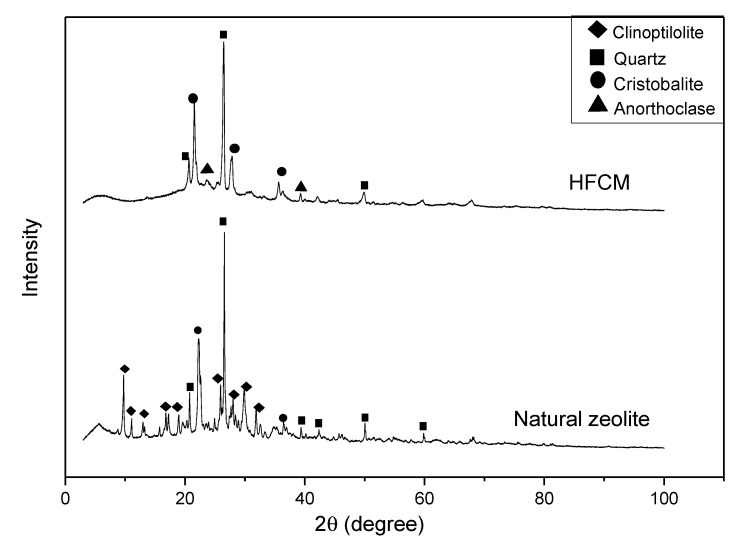
The X-ray diffraction analysis (XRD) patterns of the natural zeolite and HFCM.

**Figure 8 membranes-10-00063-f008:**
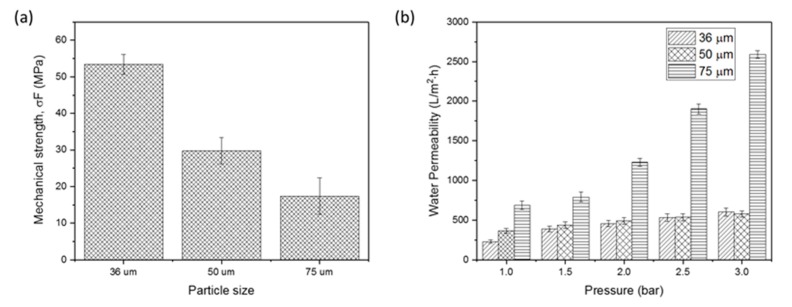
(**a**) The mechanical strength of the HFCM derived from different grade of natural zeolite particle sizes (n = 5) and (**b**) pure water permeability profile of the HFCM derived from different grade of natural zeolite particle sizes at different water pressure (n = 3).

**Figure 9 membranes-10-00063-f009:**
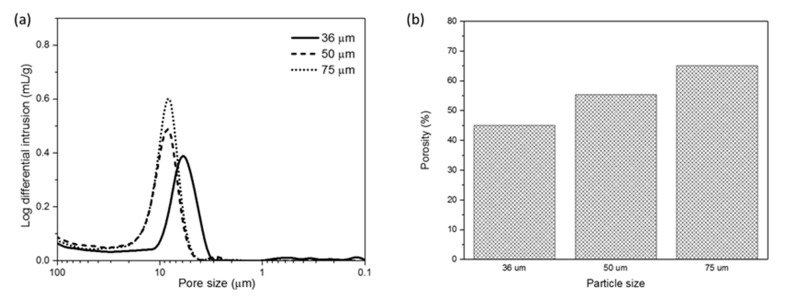
(**a**) The pore size distribution of the HFCM fabricated from different natural zeolite particle size and (**b**) porosity of the HFCM fabricated from different natural zeolite particle size.

**Figure 10 membranes-10-00063-f010:**
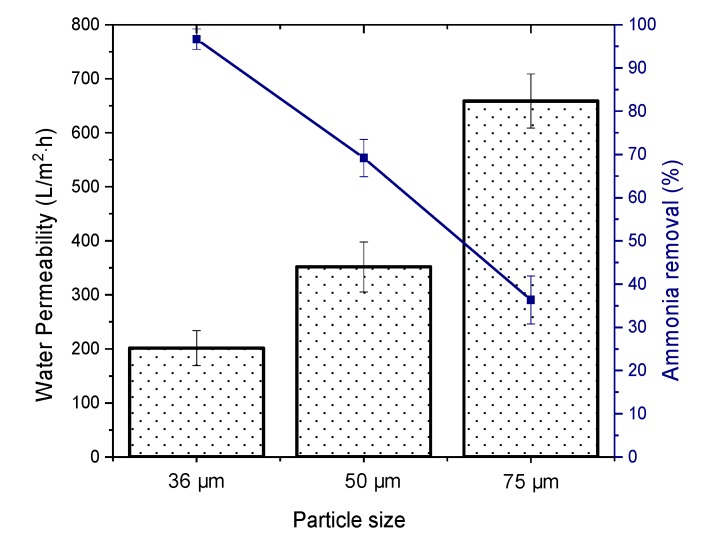
The ammonia removal and water permeability of the adsorptive HFCM derived from different particle size (n = 3).

**Figure 11 membranes-10-00063-f011:**
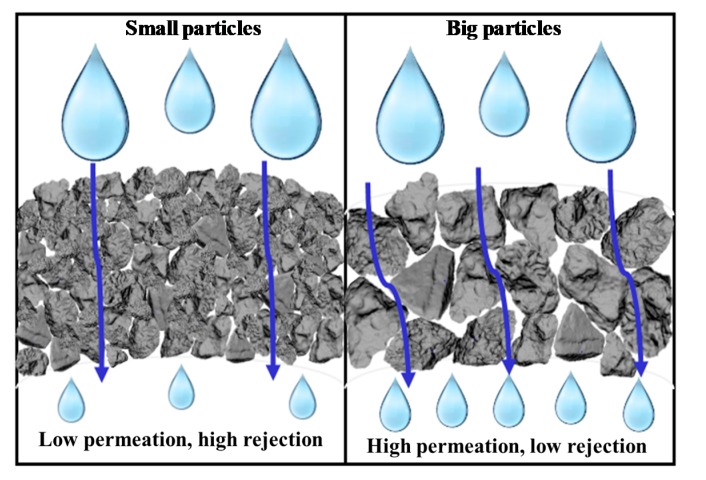
The illustration of the relationship between particle size, membrane compactness, water permeability, and ammonia rejection (adsorption).

**Figure 12 membranes-10-00063-f012:**
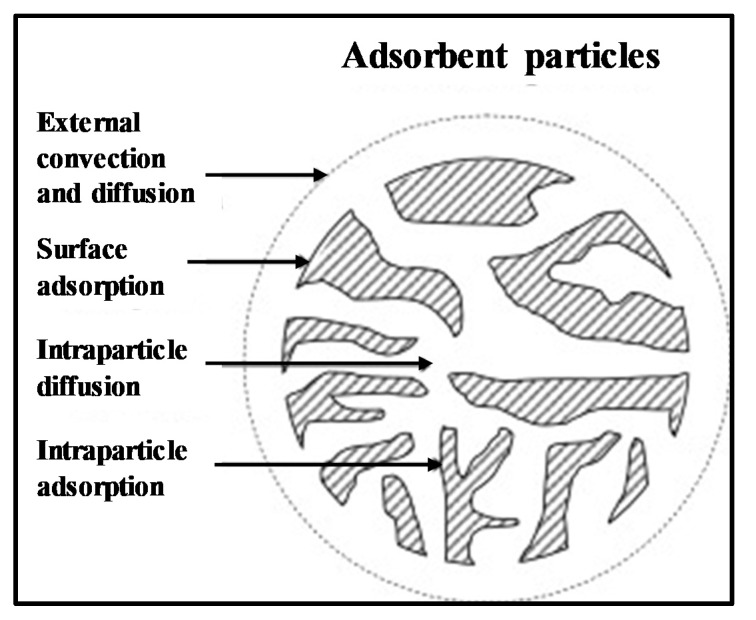
The schematic intraparticle pores of an adsorbent particle in adsorption process [51].

**Table 1 membranes-10-00063-t001:** Ammonia adsorption by natural zeolite clinoptilolites.

Material	Material Form	Initial Ammonia Concentration (mg/L)	Ammonia Removal (%)	Reference
Canadian clinoptilolite	Powder suspension	100	65	[53]
Chinese clinoptilolite	Powder suspension	115	80	[54]
Chinese Na-clinoptilolite	Powder suspension	250	77.16	[55]
Croatian clinoptilolite	Powder suspension	100	61.1	[56]
Croatian clinoptilolite	Powder suspension	800	75	[57]
Iranian clinoptilolite	Powder suspension	100	68	[58]
New Zealand clinoptilolite	Powder suspension	40	85.09	[59]
Turkish clinoptilolite	Powder suspension	150	70	[60]
USA clinoptilolite	Powder suspension	250	56	[61]
Natural zeolite clinoptilolite	Adsorptive membrane	50	82.97	This study

**Table 2 membranes-10-00063-t002:** Low-cost membranes and its applications.

Material	Fabrication Method	Sintering Temperature (°C)	Application	Membrane Mechanism	Reference
Kaolin	Extrusion	1200–1500	Arsenic removal	Membrane distillation	[62]
Chinese clay	Paste casting	400	Oil-in-water separation	Filtration	[63]
Iranian clay	Pressing	900	Cationic dye removal	Adsorption Filtration +	[64]
Kaolin + Limestone	Extrusion	800–1100	Support layer	Filtration	[65]
Indian clay	Paste casting	800–1000	Chromate removal	Flocculation + Filtration	[66]
Fly ash	Extrusion	1100–1500	Support layer	Filtration	[20]
Fly ash + bauxite	Pressing	1200–1500	Oil-in-water separation	Filtration	[67]
Kaolin + Calcium carbonate	Extrusion	1150–1300	Support layer	Filtration	[68]
Brazilian clay	Pressing	1050	Microalgae removal	Filtration	[69]
Natural zeolite clinoptilolite	Phase inversion extrusion	1050	Ammonia removal	Adsorption + Filtration	This study

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
