# Peer review of "Influence of the Natural Zeolite Particle Size Toward the Ammonia Adsorption Activity in Ceramic Hollow Fiber Membrane"

_membranes, 2020, doi:10.3390/membranes10040063_

Round 1
Reviewer 1 Report
In present manuscript, some natural zeolite-based hollow fiber ceramic membranes (HFCM) were fabricated via phase inversion method. This research mainly focused on the effect of the particle size of the natural zeolite on the physical and chemical properties of the obtained HFCMs. The topic of this manuscript is interesting, and the results and conclusions have certain reference value to related researchers. However, the authors must clarify the following problems.
- In section 2.2, the authors should provide the content of PESf in the spinning suspension.
- In line 187-205, the authors discussed the effect of the particle sizes on the shape the pore structure in detail. This discussion should be placed in section of “Morphological behavior of the HFCM”; meanwhile, the SEM images of the finger- or sponge-like membranes obtained from different zeolites should be added to support the conclusions in this part.
- In line 225-230, the discussion about the mechanical properties and the water permeation of the HFCM should be placed in “Mechanical strength and water permeability of HFCM” section.
- In line 231-243, the authors discussed the morphologies of the inner and outer surfaces of the HFCMs using SEM analysis; however, there were not any SEM images of the inner or outer surfaces in the manuscript. These SEM images should be added.
- In section 3.1.2 and section 3.1.3, the authors found that the natural zeolite edges might be melted during the sintering process, and the micropore structure of the natural zeolite may be changed, which will have a significant impact on the adsorption performance of the HFCMs. Therefore, the authors should supplement the data of the microporous structure parameters of the natural zeolite powder before and after sintering, including pore size, pore volume, porosity, specific surface area, etc.
- In section 3.2, the authors should pay more attention to the effect of the microporous structure of the HFCMs on the adsorption efficiency, not only the influence of the microporous structure among the zeolite particles.
Reviewer 2 Report
In this work, the authors studied the physical and chemical properties including membrane compactness, crystallinity, and microtopography of the membrane as well as adsorptive removal performance of ammonia using three different natural zeolite particle sizes (36, 50 and 75 μm). The results showed that the smallest particle size of 36μm was responsible for the production of the HFCM with acceptable morphology, high mechanical strength, and the lowest water permeability, ensuring a high ammonia adsorption performance of 96.67%. This work provides a new perceptiveness of the promising properties of the natural zeolite derived HFCM for ammonia removal. Some detailed suggestions are shown as follows. 1. Introduction. There are too many paragraphs in this section. The authors should introduce research background with a logical way. Preparation and application of low-cost ceramic membranes are extensively studied using various minerals and wastes such as coal fly ash, bauxite and kaolin clays. But they are not sufficiently discussed. What is the difference and novelty between this work and other existing studies? In order to give a broader background, please give a comprehensive discussion in Paragraph 7 . 2. The English writing in a more logical way should be significantly improved. Please check the accuracy of all English expressions in the text. Please correct some minor mistakes such as “Cf and Cp ” in line 159, “30s-1” in line 191, and “um” in Fig.8 (NOT LIMITTED). 3. More details such as pump and valve should be provided in the microfiltration apparatus shown in Fig.1. 4. It should be better if higher quality figures could be given. For example, the pictures of membranes and schematic of microfiltration system. For example, it is not clear to identify the magnifications in Fig.3. 5. The highlights and significance of this article should be revised. 6. What is threshold value of the alumina dope suspension viscosity in line 191 and 192? 7. There are too many repetitive descriptions between the section 3.1.1 and 3.1.2. please revise if possible. 8. I suggest the authors give a clearer explanation about the definition of “particle distribution” and its effect on ceramic membrane in section 3.1.2. 9. It is well known that the crystallinity of the same ceramic materials is only related to sintering temperature but not to particle size. So I think it doesn’t make sense to study the crystallinity properties of the HFCM with different particle size as shown in section 3.1.3. 10. A new table should be added about a comparison in materials, structure and performance and application between the membranes fabricated in the current study and other reported low-cost membranes, and discussed well in section 3.2.
Round 2
Reviewer 1 Report
The authors has improved the paper as much as possible according to the reviewer's comments. However, I still strongly recommend the author to clarify the differnent effects of micropores and macropores on the adsorption and permeability of the obtained membranes.
Reviewer 2 Report
The authors have made some efforts to improve the quality of the manuscript according to the reviewer’s comments and suggestions. How ever, there are still some concerns that are not fully and suitably addressed.
Introduction. The authors only cited three papers, some of which are not related to the background of this study, for example Ref. 15 is an adsorption-related study, not membrane. Progress of low-cost ceramic membranes such as mullite, cordierite and spinel using coal fly ash, bauxite should be extensively reviewed since they are more suitable regarding the membrane background. Actually, there is a paper published in JMS 2006, which claimed the fabrication of tubular low-cost membrane from natural zeolite (Journal of Membrane Science, 281(1-2) (2006) 592-599). It should be reviewed in the background.
The highlights of this article should be revised, focusing on the difference and significance, not research methods and results. It is good that the authors have made a new table for ammonia removal using suspensions and membrane. I still encourage the authors to add a new table saying materials, structure and performance and application between the membranes fabricated in the current study (ammonia removal) and other reported low-cost membranes (other applications such as oily water treatment). Readers will easily see what applications can be achieved for low-cost ceramic membranes.
